Comparative genomics of Pseudomonas syringae pathovar tomato reveals novel chemotaxis pathways associated with motility and plant pathogenicity

Clarke Christopher R. gtg681r@vt.edu thechrisclarke@gmail.com 1
Hayes Byron W. 1
Runde Brendan J. 1
Markel Eric 2
Swingle Bryan M. 2 3
Vinatzer Boris A. 1
1 Plant Pathology, Physiology and Weed Science, Virginia Tech , Blacksburg , VA , USA
2 Emerging Pests and Pathogens Research Unit, Robert W. Holley Center for Agriculture and Health, United States Department of Agriculture , Ithaca , NY , USA
3 Plant Pathology and Plant-Microbe Biology Section, School of Integrative Plant Science, Cornell , Ithaca , NY , USA
Chang Jeff
Electronic publication date: 2016 Oct 25
Publication date: 2016
Volume: 4
Electronic Location ID: e2570
Received 2016 Aug 5; Accepted 2016 Sep 15
Copyright: ©2016 Clarke et al.
Copyright year: 2016
Copyright holder: Clarke et al.
License: This is an open access article distributed under the terms of the Creative Commons Attribution License, which permits unrestricted use, distribution, reproduction and adaptation in any medium and for any purpose provided that it is properly attributed. For attribution, the original author(s), title, publication source (PeerJ) and either DOI or URL of the article must be cited.
License URL: https://creativecommons.org/licenses/by/4.0/

Keywords: Chemotaxis, Flagellin, Swimming motility, CheA, Swarming motility, Twitching motility, DC3000, Pto

Funding: NSF IOS-0746501 IOS-1354215 Virginia Agricultural Experiment Station Hatch Program of the National Institute of Food and Agriculture, US Department of Agriculture National Institute of Food and Agriculture postdoctoral research fellowship 2015-67012-22821 This work was supported by NSF IOS-0746501 and IOS-1354215 to BAV. Funding to BAV was also provided in part by the Virginia Agricultural Experiment Station and the Hatch Program of the National Institute of Food and Agriculture, US Department of Agriculture. CRC is supported by a National Institute of Food and Agriculture postdoctoral research fellowship (2015-67012-22821). The funders had no role in study design, data collection and analysis, decision to publish, or preparation of the manuscript.

==============================
The majority of bacterial foliar plant pathogens must invade the apoplast of host plants through points of ingress, such as stomata or wounds, to replicate to high population density and cause disease. How pathogens navigate plant surfaces to locate invasion sites remains poorly understood. Many bacteria use chemical-directed regulation of flagellar rotation, a process known as chemotaxis, to move towards favorable environmental conditions. Chemotactic sensing of the plant surface is a potential mechanism through which foliar plant pathogens home in on wounds or stomata, but chemotactic systems in foliar plant pathogens are not well characterized. Comparative genomics of the plant pathogen Pseudomonas syringae pathovar tomato (Pto) implicated annotated chemotaxis genes in the recent adaptations of one Pto lineage. We therefore characterized the chemosensory system of Pto. The Pto genome contains two primary chemotaxis gene clusters, che1 and che2. The che2 cluster is flanked by flagellar biosynthesis genes and similar to the canonical chemotaxis gene clusters of other bacteria based on sequence and synteny. Disruption of the primary phosphorelay kinase gene of the che2 cluster, cheA2, eliminated all swimming and surface motility at 21 °C but not 28 °C for Pto. The che1 cluster is located next to Type IV pili biosynthesis genes but disruption of cheA1 has no observable effect on twitching motility for Pto. Disruption of cheA2 also alters in planta fitness of the pathogen with strains lacking functional cheA2 being less fit in host plants but more fit in a non-host interaction.

Introduction

Pseudomonas syringae pv. tomato (Pto) is a common bacterial pathogen adapted to live in both agricultural and non-agricultural environments. Pto is most intensively studied for its role in causing bacterial speck disease in tomato. The Pto population is comprised of multiple closely related lineages of the pathogen. The PtoT1 lineage (which includes the well-studied eponymous member PtoT1 (Almeida et al., 2009) has dominated the population for the last 60 years in North America and Europe (Cai et al., 2011). In prior decades, the PtoJL1065 and PtoDC3000 lineages were likely the dominant field populations (Cai et al., 2011). PtoDC3000 is actually more closely related to pathogens of Brassicaceae than to PtoJL1065 and PtoT1 and its host range includes members of the Brassicaceae family (Yan et al., 2008). Strains in the PtoT1 lineage are specialists in tomato (Cai et al., 2011) but can also infect other Solanaceae (Clarke et al., 2014).

To identify the genetic features that might contribute to the recent emergence of the PtoT1 lineage, we previously sequenced and analyzed the genomes of several closely related Pto strains (Cai et al., 2011). One of the most striking non-plant-defense-related features in the genomes of PtoT1-lineage strains was the presence of several non-synonymous single nucleotide polymorphisms (SNPs) in Methyl-accepting Chemotaxis Proteins (MCPs) in Pto. We therefore hypothesized that the fine tuning of chemotaxis pathways is involved in the adaptation of Pto to its tomato host. We thus sought to identify the genetic basis for chemotaxis in Pto and characterize the importance of chemotaxis for Pto motility and interaction with plant hosts.

Many bacteria use chemotaxis pathways to control flagella-driven motility in response to environmental stimuli in a “biased random walk” (Berg & Brown, 1972). Bacteria fluctuate between moving forward (running) and reorienting (tumbling) in a controlled manner, where running is favored in the presence of increasing levels of favorable chemical cues and tumbling is favored in the presence of unfavorable chemical cues. Specific chemical cues are recognized in the periplasm by the ligand-binding domains of membrane-spanning MCPs, and signals are propagated, through a highly conserved cytoplasmic HAMP domain (Aravind & Ponting, 1999), to a histidine-aspartate phosphorelay system (see Parkinson, Hazelbauer & Falke, 2015; Wadhams & Armitage, 2004 for review). The final output is the regulation of flagellar motor rotation resulting in movement towards attractants and away from repellents. The genes involved in the two-component phosphorelay, cheA and cheY, are essential for chemotaxis in Escherichia coli (Parkinson & Houts, 1982), P. aeruginosa (Ferrández et al., 2002), and other bacteria (Porter, Wadhams & Armitage, 2011).

Chemotaxis is also linked to type IV (T4) pili-dependent motility, such as twitching motility (Kirby, 2009), in some bacteria. For example, P. aeruginosa has one chemotaxis pathway for controlling flagellar motility and a second che gene cluster involved in T4 pili formation, motility (Darzins, 1994; Whitchurch et al., 2004), and biofilm formation (Hickman, Tifrea & Harwood, 2005). Interestingly, T4 pili have previously been implicated as important in epiphytic colonization of plants (Roine et al., 1998) and have been demonstrated to be essential for virulence and surface motility by a P. syringae pv. tabaci strain (Nguyen et al., 2012; Taguchi & Ichinose, 2011). Also significant work has been done on the role of T4 pili in the insect-vectored plant pathogen Xyella fastidiosa (see De La Fuente, Burr & Hoch, 2008; Li et al., 2007 for examples) and the plant pathogen Acidovorax avenae (Bahar, Goffer & Burdman, 2009).

For plant-associated microbes, chemotaxis pathways have been best studied in diazotrophs. The α-proteobacterium Sinorhizobium meliloti, has a chemotaxis system significantly divergent from that of E. coli (Schmitt, 2002) with two cheY genes but only one cheA (Scharf, Hynes & Alexandre, 2016). CheY2 acts as the master switch for the flagellar motor like E. coli CheY (Sourjik & Schmitt, 1996), and CheY1 compensates for the lack of CheZ by acting as a phosphate sink since it can dephosphorylate CheY2 through CheA (Riepl et al., 2008). The phosphate sink regulatory mechanism of the secondary CheY proteins is also found in the α-proteobacterium Rhodobacter sphaeroides (Shah et al., 2000). In Rhizobium leguminosarum, both chemotaxis clusters contribute to motility but only one is responsible for chemotactic responses to host chemical cues in the rhizosphere (Miller et al., 2007). Also in Azospirillum brasilense motility, and specifically chemotaxis, is necessary for successful colonization of its host’s roots (Van de Broek, Lambrecht & Vanderleyden, 1998). The soil-borne close relative of P. syringae, Pseudomonas fluorescens, is also chemotactic and is attracted to several amino acid exudates of tomato roots (Oku et al., 2012).

Chemotaxis pathways are also required for optimal colonization of roots by soil-borne plant pathogens. The plant pathogens Agrobacterium tumefaciens (Hawes & Smith, 1989), Ralstonia solanacearum (Yao & Allen, 2006), and Phytophthora sojae (Morris & Ward, 1992), all rely on functional chemotaxis to effectively home in on host roots. However, chemotaxis has never been directly shown as required for plant pathogenicity after locating host roots.

In contrast to soil-borne pathogens, chemotaxis has been directly implicated in plant colonization by the foliar pathogens Xanthomonas campestris (Kamoun & Kado, 1990) and Xanthomonas citri (Moreira et al., 2015). There have been several recent advances implicating chemoperception in the interaction of P. syringae with plant hosts. Chemotaxis-associated genes were shown to be up-regulated during the epiphytic phase of invasion of the bean pathogen Pseudomonas syringae pv. syringae (Yu et al., 2013) and to play a role in vascular pathogenicity of the olive pathogen Pseudomonas syringae pv. savastanoi (Matas et al., 2012). Moreover, it has been shown that Pto swims towards open stomata of Arabidopsis thaliana leaves (Melotto et al., 2006) suggesting that P. syringae can sense some chemical cues released from stomata.

To determine the extent to which Pto employs chemotaxis and to determine its genetic basis, we characterized the chemotactic systems of Pto and elucidated the importance of chemosensory systems in regulation of bacterial motility and plant pathogenicity.

Materials and Methods

cheY phylogenetic analysis

cheY gene sequences of bacteria with previously characterized chemotaxis pathways and select additional P. syringae strains were obtained from Genbank and aligned using Megalign (DNA*, Madison, WI, USA). A neighbor joining tree was built based on this alignment using 1,000 trials and a random seed of 111. The species(strains) of bacteria included were P. syringae (PtoDC3000 (Buell et al., 2003), PtoT1 (Almeida et al., 2009), Pph1448a (Joardar et al., 2005), Psy642 (Clarke et al., 2010)), P. aeruginosa (PAO1 (Stover et al., 2000)), S. enterica (typhimurium (Stock, Koshland & Stock, 1985)), E. coli (K-12 (Blattner et al., 1997)), Rhodobacter sphaeroides (241 (Ward et al., 1995)), S. meliloti (RU11001), Bacillus subtilis (168 (Kunst et al., 1997)).

Plant and bacterial growth

Solanum lycopersicum cv. Heinz or cv. Rio Grande (tomato) seeds were sowed into 1:1 mix of promix BX (Premier Horticulture, Quebec, Candada) and metromix (Sungro, Sebe Beach, Canada) soil. A. thaliana ecotype Columbia seeds were stratified for 3 days in water at 4 °C and then sowed into Sunshine #1 (Sungro, Sebe Beach, Canada) soil. All plants were grown for 4–5 weeks under a laboratory growth light shelf at 22 °C and 12-hour light cycles.

All bacteria were grown overnight at 28 °C on King’s B (KB, King, Ward & Raney, 1954) plates with 1.5% agar and 25 µg/ml tetracycline (all strains included the empty vector pme6010 to use tetracycline as an antibiotic marker) before use in assays. For measuring growth of strains in liquid culture, bacteria were diluted in 10 mM MgSO4 to an optical density at 600 nm wavelength (OD600) of 0.01. 5 µL was added to 5 ml a test tube of either liquid KB media or liquid Minimal Media (MM) (Huynh et al., 1989) and placed in a 28 °C shaking incubator. 10 µL of the media was removed from the tubes at the indicated time points, diluted, and then plated on KB-tetracycline plates. Plates were incubated at 28 °C, the number of colony forming units were counted, and the number of CFUs/ml in the test tube at the sample time was calculated.

Swim and swarm plates

Swim and swarm plates were made by making standard KB media plates with the indicated agar concentrations instead of the standard 1.5% agar concentration and adding tetracycline to 25 µg/ml. Swim and swarm plates were always used 4–5 h after they were made. 2 µL of bacteria diluted in 10 mM MgSO4 to an OD600 of 0.01 were pipetted onto the plates, with 3 bacteria strains/plate. Strains being directly compared were inoculated onto the same set of plates to account for plate-to-plate variability. 10 min after the inoculation, the lid of the plate was lightly sprayed with water and the plate was flipped upside down into the lid (so that the wet inside of the lid is at the bottom, followed by an air gap, followed by the bacteria on the agar media at the top) and sealed with parafilm. Maximum cross section of the colony spread was measured after a two-day incubation at 28 °C or 21 °C. In these plates, if a strain is either non-motile or unable to tumble to change directions the bacteria cannot spread beyond the point of inoculation. Fully motile and chemotactic bacteria spread on the plate due to local depletion of nutrients leading to a nutrient gradient and chemotactically driven swimming motility toward local regions with more nutrients.

Split capillary assay

Capillary assays were modified from Adler (1973). A ring of grease was created on a glass coverslip. Bacteria diluted in 10 mM MgSO4 to an OD600 of 0.01 were pipetted into the grease ring to form a pool of the bacteria. One 1 µL capillary tube (Drummond Scientific, Broomall, PA, USA) was filled with 10 mM MgSO4, sealed at one end with parafilm, and inserted at the open end into the pool of bacteria. A second capillary tube was filled with KB media, sealed at one end with parafilm, and inserted at the open end into the pool of bacteria. Extra grease was placed on top of the capillary tubes where they contact the grease ring and the pool was sealed with a coverslip on the top (see Fig. S6). The coverslip sandwich was left undisturbed for 45 min. Following the 45-minute incubation the contents of the capillary tube were diluted, plated onto solid KB-tetracycline plates, and incubated at 28 °C for two days The number of colony forming units (CFUs) originating from each capillary tube was counted and used to calculate the ratio of the number of CFUs from the KB-containing capillary over the number of CFUs from the matching 10 mM MgSO4 capillary.

Creation of chemotaxis disruption and deletion mutants and molecular cloning of chemotaxis genes

Genome disruptions of the cheA1 and cheA2 genes were created via the P. syringae gene disruption construct pBAV208 using a previously described approach (Clarke et al., 2010) and the primers listed in Table S2. The disruptions result in strains with two fragments of the cheA genes. The cheA1 disruption mutants have a 5′ cheA1 fragment with an in-frame stop codon at position 261 and a 3′ fragment starting with a stop codon. The cheA2 disruption mutants have a 5′ cheA2 fragment with an in-frame stop codon at position 281 and a 3′ fragment starting with a stop codon. Plasmids were conjugated into PtoDC3000 and Pto1108 via triparental mating. Major results were confirmed with second, independent disruption mutants of cheA1 and cheA2 in both PtoDC3000 and Pto1108. Disruption mutants are designated as either cheA1− cheA2− strains throughout this paper.

The PtoDC3000 ΔcheA1, ΔcheA2 and ΔcheA1cheA2 deletion mutant strains were constructed using the recombineering methods described in Swingle et al. (2010) and Bao, Cartinhour & Swingle (2012). The ΔcheA1 mutant was constructed by transforming PtoDC3000 containing pUCP24/recTE with a recombineering substrate designed to replace the cheA1 gene with the kanamycin resistance encoding neo gene flanked by modified frt sequences (frt-neo-frt). The cheA1 deletion recombineering substrate was amplified by PCR using primers oSWC06647 and oSWC06648 and pKD4 as a template. This product contained the frt-neo-frt cassette flanked by 80 bp sequences homologous to PtoDC3000 genome coordinates 996501-996580 and 994354-994433 at the left and right end, respectively. Kanamycin resistant recombinants were selected and confirmed to contain the frt-neo-frt cassette in the correct location by PCR. The cheA1 deletion recombinants were then transformed with pCPP5264, which expresses the FLP recombinase and catalyzes site-specific recombination between frt sequences to remove the neo gene. The neo gene was confirmed to be deleted by PCR and the recombinant strains were confirmed to have lost the pUCP24/recTE and pCPP5264 plasmids. The structure of the mutant was confirmed by sequence analysis to consist of the first 6 codons of the cheA1 gene, fused in frame to the 28 codon frt scar and followed by 6 terminal codons of the cheA1 gene.

The cheA2 deletion strains were then constructed using recombineering to introduce the mutation into wild-type and ΔcheA1 backgrounds to yield the cheA2 and cheA1cheA2 deletion strains. The cheA2 recombineering substrate was generated using long flank homology PCR as described in Swingle et al. (2010). The cheA2 recombineering substrate was composed of the frt-neo-frt cassette with a 516 bp right flank and 556 bp left flank homologous to PtoDC3000 genome coordinates 2166604-2167120 and 2169335-2169890. The cheA2 deletion recombineering substrate was used to transform wild-type and cheA1 strains containing the pUCP24/recTE recombineering plasmid; recombinants were selected for resistance to kanamycin. The integration of the frt-neo-frt deletion cassette at the correct location was confirmed by PCR. These strains were then transformed with pCPP5264 to catalyze the excision of the neo gene. PCR was used to demonstrate that the neo gene had been deleted and the pUCP24/recTE pCPP5264 plasmid was cured from the cheA2 deletion strains. The final structure of the deletion mutants was confirmed by sequencing to consist of the first 6 codons of the cheA2 gene fused in frame to the frt scar and the terminal six codons of cheA2.

For the complementation strains, cheA1 and cheA2 were individually cloned into the P. syringae expression vector pme6010 using a previously described approach (Clarke et al., 2013) under control of the constitutive npt2 promotor and the primers listed in Table S2. cheA1 was cloned including 25 bp upstream of the start codon and cheA2 was cloned including 14 bp upstream of the start codon. The pme6010 plasmids containing cheA1 and cheA2 were conjugated into PtoDC3000 and Pto1108 wild type and cheA1/cheA2 disruption/deletion strains via triparental mating.

Plant infection assays

Plant infections were carried out under a laboratory growth shelf (12 h light cycle) as previously described (Clarke et al., 2013). Briefly, spray infections were performed with 0.01 OD600 of freshly grown bacteria on 4- or 5-week-old tomato or A. thaliana plants 24 h after the plants were sprayed with water and placed under a humidity dome. High humidity was maintained for 16 h following infection and leaves were sampled 4 days post infection using a 4 mm cork borer for quantifying total bacterial growth (both endophytic and epiphytic populations) as previously described (Clarke et al., 2013) using KB-tetracycline plates.

Results

Single nucleotide polymorphisms in a recently emerged Pto lineage are enriched in chemotaxis-associated genes

The genome sequences of the extremely closely related strains within the T1 lineage of Pto were previously compared to identify single nucleotide polymorphisms (SNPs) as candidates for the recent success of the PtoT1 lineage in tomato field populations in the past 50 years (Cai et al., 2011). Only 265 SNPs are present among the genomes of these strains (Cai et al., 2011). Seven non-synonymous SNPs were in the coding sequence of putative MCPs. This enrichment of SNPs in MCPs, suggests that chemo-detection systems are involved in the adaptation of the Pto lineage on tomato. Six of the seven non-synonymous SNPs are in the periplasmic domain of the MCPs (Fig. S1), which is the domain responsible for recognizing specific chemoattractants/repellants (Parkinson, Hazelbauer & Falke, 2015). This pattern suggests that adaption in recognition of chemical compounds in the Pto lineage is potentially contributing to the recent clonal expansion of the PtoT1 lineage. We therefore proceeded to characterize the chemosensory system of Pto in both the model strain PtoDC3000 and a genetically-tractable representative of the PtoT1 lineage in which the SNPs were identified, strain PtoNCPPB1108 (Pto1108 for short, Table 1).

Table 1 Strains used in this study.

Strain	Plasmid	Description	Source	
Pto1108	6010:empty	wild type	This work	
PtoDC3000	6010:empty	wild type	This work	
Pto1108 cheA1−	6010:empty	cheA1 disruption mutant	This work	
Pto1108 cheA2−	6010:empty	cheA2 disruption mutant	This work	
Pto1108 cheA1− (comp)	6010:cheA1	cheA1 disruption mutant (complemented)	This work	
Pto1108 cheA2− (comp)	6010:cheA2	cheA2 disruption mutant (complemented)	This work	
PtoDC3000 cheA1−	6010:empty	cheA1 disruption mutant	This work	
PtoDC3000 cheA2−	6010:empty	cheA2 disruption mutant	This work	
PtoDC3000 cheA1− (comp)	6010:cheA1	cheA1 disruption mutant (complemented)	This work	
PtoDC3000 cheA2− (comp)	6010:cheA2	cheA2 disruption mutant (complemented)	This work	
PtoDC3000 ΔcheA1	6010:empty	cheA1 deletion mutant	This work	
PtoDC3000 ΔcheA2	6010:empty	cheA2 deletion mutant	This work	
PtoDC3000 ΔcheA1 (comp)	6010:cheA1	cheA1 deletion mutant (complemented)	This work	
PtoDC3000 ΔcheA2 (comp)	6010:cheA2	cheA2 deletion mutant (complemented)	This work	
PtoDC3000 ΔfliC	6010:empty	fliC deletion mutant	Clarke et al. (2013)	
PtoDC3000 ΔpilA	6010:empty	pilA deletion mutant	Roine et al. (1998)	

The Pto genome contains two primary chemotaxis gene clusters

The previously sequenced Pto genomes (Buell et al., 2003; Cai et al., 2011) all have two gene clusters with canonical cheA-cheY two-component phosphorelays and three other clusters of putative chemotaxis-associated genes but lacking the histidine kinase cheA and response regulator cheY genes (Fig. 1A, Table S1). The che1 cluster is neighbored by genes associated with pili biosynthesis and syntenically similar to the che2 cluster in P. aeruginosa (Kato et al., 2008). The che2 cluster is syntenically similar to the che clusters of E. coli and P. aeruginosa (Kato et al., 2008) and immediately downstream of flagellar-biosynthesis genes like in the genomes of many other gram-negative bacteria.

Figure 1 Chemotaxis gene clusters in Pto.

The genome of Pto1108 contains multiple chemotaxis gene clusters. (A) The organization of the chemotaxis gene clusters in the genome of Pto1108. (B) Neighbor-joining tree based on aligned CheY protein sequences from bacteria with previously characterized chemotaxis pathways and select other P. syringae strains. The full species and strain names are listed in the methods. Numbers at nodes represent bootstrap support based on 1,000 trials.

Figure 2 PtoDC3000: cheA2− strains are deficient in chemotactic swimming motility.

(A) Example pictures and box plots of the colony diameter two days after inoculation of the indicated strains on 0.28% agar KB swim plates. (B) The ratio of colony forming units of the indicated bacteria that entered a capillary tube of KB media over a capillary tube of 10 mM MgSO4 in the split capillary assay. Asterisks indicate statistical significance compared to wild type in a Student’s t-test at the indicated p-values. Data represent the average of eight replicates and error bars are the standard error. Essentially identical results were obtained in at least three independent experiments for all strains.

Phylogenetic analysis of cheY gene sequences revealed that Pto cheY2 clusters with high support (bootstrap = 100) with cheY genes known to be essential for flagellar regulation in other gammaproteobacteria (Fig. 1B). Pto cheY1 clusters with cheY genes not associated with flagellar motility in other bacteria. We therefore hypothesized that the Pto che2 pathway is the canonical chemotaxis pathway regulating flagellar switching and the che1 pathway has a distinct role, potentially functioning in regulation of pili-based motility.

The Pto genome encodes three additional non-canonical chemotaxis gene clusters. Like che2, the che3 cluster is also flanked by flagellar biosynthesis genes. The che4 and che5 clusters each contain a putative non-canonical histidine kinase–response regulator two-component system, as well as cheB and cheR, which encode receptor-modifying enzymes, and cheW, which codes for an adaptor protein (Fig. 1A). The Pto genome encodes 48 annotated MCPs in total.

The che2 pathway in Pto regulates swimming motility

To assess the importance of the two major chemotaxis gene clusters, we created disruptions in the main signal transduction genes of the che1 and che2 clusters, cheA1 and cheA2, individually in PtoDC3000 and Pto1108. The disruption mutants are referenced as cheA1− and cheA2− throughout the manuscript and figures. We also created in-frame gene deletions of cheA1 and cheA2 in PtoDC3000. The deletion mutants are referenced as ΔcheA1 and ΔcheA2. We quantified swimming motility using low-agar-concentration (0.28%) KB swim plates that quantify flagellar-based motility and chemotactic function (see methods). cheA2 was essential for motility of both PtoDC3000 and Pto1108 in the swim plates (Fig. 2A, Fig. S2A) and phenotypically identical to the fliC deletion mutant of PtoDC3000. The same phenotypes were observed with second, independent disruption mutants of cheA1 and cheA2 in both the PtoDC3000 and Pto1108 background. Complementation of cheA2 in the PtoDC300cheA2− background restored swimming motility, but not to the level of the wild type strain (Fig. 2A), potentially because the disruption insert was polar leading to misregulation of other genes in the che2 cluster or non-optimized expression of cheA2 (See Fig. 1A and Table S1).

To determine whether cheA2 is essential for motility or only chemotactic regulation of motility, the swimming behavior of the strains were observed in liquid KB media using dark-field microscopy at 400x magnification. Both Pto1108cheA2− and PtoDC3000 cheA2− exhibited a “smooth-swimming” phenotype—motile, but unable to tumble to change swimming direction. Pto1108cheA1- and PtoDC3000cheA1− both swam and tumbled similar to wild type strains (Videos S1–S3). Flagellar mutants, in contrast to the cheA2 mutants, are completely non-motile in this assay.

Additionally, in a variant of the classic capillary assay (Adler, 1973) which tests chemotactic function based on the ability of bacterial cells to preferentially move into a nutrient-rich medium, cheA2 was necessary for full chemotactic function in PtoDC3000 (Fig. 2B). The cheA2− dependent aberrations in these assays are indicative of loss of directional control of swimming motility and not general defects in growth, because the PtoDC3000 and Pto1108 wild type and chemotaxis disruption mutant strains replicate at equivalent rates in both liquid plant-apoplast-mimicking Minimal Media (MM) and rich KB media (Fig. 3, Fig. S3A).

Figure 3 Surface motility in Pto and chemotaxis mutants.

Neither cheA1 nor cheA2 is required for optimal growth of PtoDC3000 in liquid KB media. PtoDC3000, PtoDC3000 cheA2−, and PtoDC3000 cheA1-were grown in liquid KB (A) and minimal media (B). ns = not significantly different from wildtype in a Student’s t-test at p < 0.05. Data represent the average of four replicates and error bars are the standard error. Essentially identical results were obtained in 2 independent experiments.

Swim plate motility was also eliminated in the PtoDC3000 ΔcheA2 deletion mutant and mostly rescued by ectopic expression of cheA2 (Fig. S2B). The PtoDC3000 ΔcheA1 deletion mutant was also partially impaired in swimming motility on swim plates, but complementation of cheA1 did not rescue the swimming defect (Fig. S2B). PtoDC3000 ΔcheA1 grew slower than wild type in liquid culture (Fig. S3B) suggesting a general growth defect in this strain, potentially due to changes in the duplication state of an unstable region in the PtoDC3000 genome (Bao et al., 2014). We therefore conclude that mutations in cheA2 but not cheA1 compromise regulation of the flagellar motor in both PtoDC3000 and Pto1108, demonstrating that the che2 cluster is the primary cluster responsible for controlling flagellar-mediated chemotaxis. Because of the observed growth defect in the chemotaxis deletion mutants, we primarily relied on the disruption mutants in the subsequent assays.

Type 4 (T4) pili-regulated twitching motility is not controlled by the che1 pathway in Pto

Because the che1 gene cluster is flanked by a gene cluster annotated to encode for components of T4 pili, we hypothesized that the che1 cluster might play a role in chemotactic control of T4 pili similar to the che2 cluster of P. aeruginosa (Whitchurch et al., 2004). To test this hypothesis, we employed KB plates with different agar concentrations (0.4–1.3%) that allow for the observation of surface motility. We quantified surface motility by inoculating these plates with wild type PtoDC3000 and the PtoDC3000 cheA2− and PtoDC3000 cheA1− disruption mutants. We also inoculated the surface motility plates with PtoDC3000 ΔfliC (Clarke et al., 2013) and PtoDC3000 ΔpilA (a T4 pili-deficient deletion mutant, (Roine et al., 1998)) as controls for strains deficient in surface swarming and twitching motility respectively. PtoDC3000 ΔpilA is mostly non-motile on these plates (Fig. 4A), similar to previous observations (Roine et al., 1998), though would occasionally expand slightly beyond the inoculation site. PtoDC3000 ΔfliC and PtoDC3000 cheA2− were both motile starting at 0.6% agar concentration (Figs. 4A and 4B). PtoDC3000 cheA1− is fully motile at all agar concentrations revealing that cheA1 is not required for surface motility (Fig. 4B). Similar phenotypes were observed with the Pto1108 chemotaxis disruption mutants except Pto1108 is unable to move effectively on high concentration agar (>1.2%) (Fig. S4A). The same phenotypes were observed with second, independent disruption mutants of cheA1 and cheA2 in both the PtoDC3000 and Pto1108 background.

Figure 4 Effect of temperature of surface motility of Pto.

Neither cheA1 nor cheA2 is required for surface motility at 28 °C. Data represent the average of seven replicates and error bars are the standard error. * indicates significant differences in swim diameter for any strain between the two temperatures at the indicated p-values using a Student’s t-test. Essentially identical results were obtained in at least two independent experiments for all strains at 0.4, 0.5, 0.6, 0.7, 0.9, 1.1 and 1.3% agar concentrations.

Figure 5 Both pilA-dependent and cheA2/fliC-dependent surface motility in PtoDC3000 are thermo-regulated.

Surface motility plate assays using 0.5% and 0.9% agar were performed with the PtoDC3000 chemotaxis mutants (A and C) and the motility mutants (B and D) at both 28 °C and 21 °C. Data represent the average of eight replicates and error bars are the standard error. * indicates significant differences in swarm diameter for any strain between the two temperatures at the indicated p-values using a Student’s t-test. Essentially identical results were obtained in at least 3 independent experiments for all strains and all temperature/agar percentage combinations.

Surface but not swimming motility is temperature-dependent in Pto

Production of surfactants, flagella components and other products required for motility by P. syringae pv. syringae are thermo-regulated with expression reduced at temperatures greater than 25 °C and completely repressed at 30 °C (Hockett, Burch & Lindow, 2013). We therefore tested both swimming and surface motility of all chemotaxis and motility mutant strains at both 21 °C and 28 °C to ascertain whether our previously observed motility phenotypes were affected by the temperature they were originally performed (28 °C). All of the strains spread more slowly on swim plates at 21 °C, which is closer to the optimal temperature for swimming motility (Cuppels, 1988), than 28 °C but had no effect on the phenotype of any mutant relative to wild type (Fig. S5).

Conversely, the effects of knocking out chemotaxis and motility genes in Pto on surface motility was significantly temperature-dependent. PtoDC3000 ΔpilA was motile on both 0.5% and 0.9% agar at 21 °C but not at 28 °C (Figs. 5 and 5B), suggesting a pilA-independent motility mechanism in Pto for motility on semi-solid surfaces that is repressed at higher temperatures. PtoDC3000 ΔfliC and PtoDC3000 cheA2− were only able to spread at 28 °C but not 21 °C (Figs. 5C and 5D). Again, cheA1 was not essential for surface motility under any conditions (Figs. 5C and 5D). Moreover, cheA1 was not essential for surface motility in strain Pto1108 at any temperature tested (Fig. S4B). Pto1108cheA2− was also motile on higher agar concentrations (0.9%) at 28 °C but not 21 °C (Fig. S4B) suggesting temperature regulation of swarming motility in this strain as well.

Both chemotaxis pathways are required for full in planta fitness of Pto

To test the importance of chemotaxis during plant-Pto interactions, tomato plants (Solanum lycopersicum cv. Heinz) were spray inoculated with either wild type or chemotaxis disruption mutant strains of Pto. Total in planta bacterial population sizes were quantified 4 days post inoculation. Both chemotaxis pathways are necessary for full in planta fitness of both PtoDC3000 and Pto1108 (Fig. 6A), and cheA2 is essential for full pathogenicity of Pto1108 in tomato (Fig. 6B), though there was substantial variability within and among independent experiments potentially reflecting small differences in humidity or other environmental conditions. Additionally, both chemotaxis mutants of PtoDC3000 have reduced fitness on A. thaliana (another plant host of PtoDC3000, Fig. 6C), suggesting that pathogen chemotaxis is an important factor in multiple plant-microbe interactions. This phenotype was confirmed with independent disruption mutants for all strain-plant combinations. The reduced growth is not due to general fitness defects as the chemotaxis disruption mutants grow as well as the wild type strain in liquid culture (Fig. 3, Fig. S3A). Neither cheA1 nor cheA2 was essential for pathogenicity when inoculated via infiltration directly into the apoplast of A. thaliana or tomato (Fig. S6). We therefore conclude that the chemotaxis pathways are primarily functioning during the epiphytic phase of Pto plant infection. All plant infections were confirmed at least twice with independent cheA disruption mutants, but ectopic expression of cheA1 or cheA2 was insufficient to consistently rescue plant pathogenicity.

Figure 6 Disruptions in either the che1 or che2 pathway affect plant pathogenicity of Pto.

(A–D) The population density of strain PtoDC3000 (A and C) or strain Pto1108 (B and D) four days following spray inoculation of the indicated plants. Data represent the average of six replicates and error bars are the standard error. Asterisks represent significant difference in a Student’s t-test between each mutant and the corresponding wild type strain (*, p < 0.05, **, p < 0.01). The fraction of independent experiments resulting in at least a 5-fold difference in growth relative to the wild type strain are shown at the bottom of the bar for each mutant strain.

Disruption of the che2 pathway increases the fitness of Pto strain 1108 on the non-host pathogen A. thaliana

In contrast to the attenuated growth of the chemotaxis mutants on susceptible plants, Pto1108 cheA2− grew to significantly higher population densities than wild type Pto1108 on A. thaliana, a non-host plant for Pto1108 (Fig. 6D). This result indicates that functional chemotactic systems contribute to the resistance phenotype in this non-host interaction.

Discussion

Mutations in chemosensory systems underscore recent clonal shifts in field populations of Pto

The worldwide field population of Pto has undergone a significant population shift with the PtoT1 lineage becoming the dominant clone over the past 60 years (Cai et al., 2011). Comparisons between the genomes of Pto1108, an early PtoT1 strain, and several more recent PtoT1 strains revealed that several putative chemotaxis-associated genes are under selection in the now dominant PtoT1 lineage. This pattern suggests that changes in chemotactic systems may be adaptations underpinning the Pto population shift. Before testing this hypothesis, it was necessary to first test the broader hypothesis that chemotaxis pathways are functional in—and important for—Pto during its lifecycle.

The che2 pathway, but not the che1 pathway, is required for multiple Pto motility mechanisms

We identified multiple chemotaxis clusters in the Pto genome (Fig. 1) and tentatively proposed that the che2 cluster encodes the canonical flagella-controlling chemotaxis pathway based on sequence and syntenic similarity to chemotaxis pathways in other gram-negative bacteria. All tested cheA2 disruption and deletion mutants were phenotypically identical to the flagella-minus fliC mutant in swim plates, split capillary assays, and surface motility assays (Figs. 2 and 4). We therefore conclude that the che2 pathway is the canonical chemotaxis pathway in Pto controlling flagellar motility.

The function of the che1 pathway in Pto remains a mystery. We had hypothesized that the che1 pathway was controlling pili-dependent twitching motility because of its sequence and syntenic similarity to the pili-controlling chemotaxis cluster in P. aeruginosa (Whitchurch et al., 2004) and its genomic position next to pili biosynthesis genes (Fig. 1). However, this hypothesis was not supported by our data because the cheA1 mutants behaved identically to the wild type Pto strains in surface motility (Fig. 4). The PtoDC3000 ΔpilA strain did, as expected, exhibit aberrant surface motility behavior.

Pto has multiple temperature-dependent surface motility mechanisms based on the divergent phenotypes observed at 28 °C compared to 21 °C. Unlike P. syringae pv. syringae (Hockett, Burch & Lindow, 2013), surface motility of wild type Pto was not markedly affected at 28 °C compared to 21 °C. However, putative swarming motility was likely downregulated at 28 °C but was compensated for by twitching motility in Pto. Specifically, we found that Pto has an additional fliC- and cheA2-dependent surface motility mechanism as previously shown (Nogales et al., 2015), which is active only at higher temperatures. pilA was essential for surface motility at 28 °C and fliC and cheA2 were essential for surface motility at 21 °C (Fig. 5) revealing that Pto has at least two genetically distinct mechanisms for surface motility, both of which are cheA1-independent. These results suggest that swarming motility is favored at lower temperatures and twitching motility favored at higher temperatures for Pto. The nature of these distinct mechanisms and how Pto switches from a fliC/cheA2-dependent to a pilA-dependent motility mechanism as temperatures increase remains to be elucidated.

Pto requires functional chemotaxis for optimal plant pathogenicity

P. syringae strains, including Pto, can live in myriad environments but are most intensively studied for their role as the causative agents of plant disease. The identified chemotaxis pathways are potentially used in numerous phases of the Pto lifecycle. In this work we establish that fitness of Pto on host plants is potentially dependent on both the che2 and che1 pathways (Fig. 6) though high experiment-to-experiment variability remains an issue. The function of the che1 pathway remains unknown, precluding speculation about the mechanism by which mutations in cheA1 reduce the fitness of Pto in plants. The primary role of the che2 pathway appears to be regulating rotational bias of the flagellar motor and we presume that the primary cause of the fitness defect associated with mutations in cheA2 in Pto is a result of the loss of flagella-dependent motility. However, in previous work we established that the PtoDC3000ΔfliC strain is not required for optimal pathogenicity of plants following spray inoculation (Clarke et al., 2013). It is therefore challenging to interpret the finding that cheA2 mutants are less fit on plant hosts. We propose that either (1) the che2 pathway is required by Pto for functions other than flagellar motor control during plant infections, or (2) the wild type-level pathogenicity of the PtoDC3000ΔfliC strain on tomato is the result of a counterbalance between a decrease in pathogenicity due to loss of flagella function and an increase in pathogenicity due to loss of several flagellin-derived elicitors of plant immunity (Clarke et al., 2013).

This conclusion warrants caution because ectopic expression of cheA did not rescue the pathogenicity of the cheA disruption mutants and experiment-to-experiment variability. We hypothesize that complementation is not successful in this case to rescue the pathogenicity because of potential polar effects on genes in the che clusters downstream of cheA. This hypothesis is supported by the observation that ectopic expression of cheA2 in the PtoDC3000 cheA2− strain only partially restored swimming motility (Fig. 2A). Though ectopic expression of cheA2 fully rescued swimming motility in the PtoDC3000 ΔcheA2 strain, we were unable to use the deletion mutants in the plant pathogenicity assays because of a general growth defect in these strains (Fig. S3B).

Regarding, the variability of the severity of attenuation of plant pathogenicity of cheA1 and cheA2 mutants, we propose that the effect is dependent on specific environmental conditions (such as humidity, daytime, and temperature (Hirano & Upper, 2000; Wilson, Hirano & Lindow, 1999)). Our finding that Pto alters its predominant mechanism of surface motility based on temperature (Fig. 5) supports the proposition of environmental conditions playing a crucial role in determining plant pathogenicity. The optimal growth conditions for Pto to use chemotaxis to maximize plant pathogenicity remain to be determined. It is important to note that alterations in in planta fitness of the disruption mutants was confirmed using second independent disruption mutants. Additionally, differences were only observed in one direction; no experiments resulted in the opposite phenotype shown in Fig. 6. Finally, both cheA1 and cheA2 mutants were only essential for pathogenicity following spray-inoculation, not infiltration-inoculation. We therefore propose that Pto is primarily using its chemosensory system during the epiphytic phase of plant infection that is bypassed during infiltration-inoculation. Future experiments to distinguish epiphytic vs. endophytic growth of Pto and the chemotaxis mutants will help clarify this possibility.

Functional chemotaxis pathways are detrimental to Pto1108 in a non-host interaction

Surprisingly, Pto1108 cheA2− was a more successful pathogen than wild type Pto1108 on the non-host plant Arabidopsis (Fig. 6D), though again we observed significant experiment-to-experiment variability. However, it is worth noting that Pto1108 cheA2− grew better or the same as wild type Pto1108 in all experiments and never worse than wild type. We hypothesize that this increase in pathogenicity is a result of Pto1108 cheA2− strain triggering a weaker immune response in A. thaliana than wild type Pto1108. Specifically, we propose that Pto1108 cheA2− triggers fewer A. thaliana defenses, because it has an extended epiphytic phase avoiding detection by the plant immune system. In this model, loss of chemotactic control of the flagellar motor results in the strain being unable to locate stomata or other openings into the apoplast. This inability to switch from an epiphytic to an endophytic lifestyle is harmful for strains on host plants because they are equipped to avoid and suppress the plant immune system while invading the nutrient rich apoplast and escaping UV and desiccation stress on the leaf surface (Wilson, Hirano & Lindow, 1999) and therefore benefit from becoming endophytes. Alternatively, during infection of non-host plants, the microbe benefits from remaining epiphytic, because it is ill-equipped to suppress the strong plant immune responses activated during endophytic invasion. Experimental evidence for both the attenuated elicitation of plant immune responses and extended epiphytic lifestyle of cheA2 mutants will greatly strengthen confidence in this model.

Conclusions

These results demonstrate the importance of the chemotactic systems of Pto for bacterial motility and pathogenicity in plants. We identified and characterized the che2 cluster as the chemotaxis cluster that regulates flagellar-dependent swimming motility and swarming surface motility. Surface motility of Pto is likely thermo-regulated with swarming motility favored at low temperatures (21 °C) and twitching motility favored at higher temperatures (28 °C). The che2 cluster is also essential for optimized pathogenicity of Pto1108 and PtoDC3000 on plant hosts, potentially during the epiphytic phase of plant invasion. The che1 cluster also plays a potential role in PtoDC3000 pathogenicity of tomato though the role of che1 in motility remains unresolved.

Building upon this foundation, it will be possible to exploit the natural variation in chemotaxis genes to discover if chemosensory systems contribute to the host range and adaptation of Pto strains and other bacterial plant pathogens. Specifically, future work can address the hypothesis that the seven identified non-synonymous SNPs in MCPs contribute to improved fitness of the recent PtoT1 strains in tomato field populations.

Supplemental Information

Video S1 Swimming motility of Pto1108 in KB rich media

Click here for additional data file.

Video S2 Swimming motility of Pto1108 cheA1− in KB rich media

Click here for additional data file.

Video S3 Swimming motility of Pto1108 cheA2− in KB rich media

Click here for additional data file.

Supplemental Information 1 Supplementary Information

Click here for additional data file.

Data S1 Raw Data

Compressed folder containing all of the raw data associated with figures 2(bacterial capillary assay plate counts and swim plate measurements), 3(bacterial plate counts from liquid media growth), 4(surface plate measurements), 5(surface plate measurements), 6(bacterial plate counts from in planta growth), S2(swim plate measurements), S3(bacterial plate counts from liquid media growth), S4(surface plate measurments) and S5(swim plate measurements) along with associated metadata.

Click here for additional data file.

We sincerely thank Ben Webb (VT) for technical guidance, Joshua Clarke for assistance with plotting in Matlab, and Birgit Scharf (VT) for technical guidance and critical review of the manuscript.

Additional Information and Declarations

Competing Interests

Author Contributions

Data Availability

The authors declare there are no competing interests.

Christopher R. Clarke conceived and designed the experiments, performed the experiments, analyzed the data, wrote the paper, prepared figures and/or tables, reviewed drafts of the paper.

Byron W. Hayes and Brendan J. Runde performed the experiments, reviewed drafts of the paper.

Eric Markel and Bryan M. Swingle contributed reagents/materials/analysis tools, reviewed drafts of the paper.

Boris A. Vinatzer conceived and designed the experiments, reviewed drafts of the paper.

The following information was supplied regarding data availability:

The raw data has been supplied as a Supplementary File.

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
