# Peer review of "Comparative genomics of Pseudomonas syringae pathovar tomato reveals novel chemotaxis pathways associated with motility and plant pathogenicity"

_PeerJ, doi:10.7717/peerj.2570_

## Round 0.1 · original submission · Major Revisions

Due to the lengthy comments, I have also attached my comments in PDF form.

The two reviewers have provided some very insightful comments that I ask you to address. In general, the reviewers’ suggestions are centered on three main themes:

1) data that reveal insights into the proposed polar nature of the insertion mutants, e.g. qRT-PCR of neighboring genes, and data showing the complementing clone was sufficiently expressed, e.g., qRT-PCR or western of a tagged protein.

2) significant improvements in data presentation/analysis and greater transparency in experimental design (conclusions should recognize limitations of the experiments). In particular, I agree with the reviewers that it is difficult to draw conclusions regarding in planta behavior in the absence of key information. Please also recognize that there is a potential for the temperature (and perhaps fluctuations, in environments that are not as controlled as a growth chamber) in which plants were grown and inoculated, to have affected the phenotypes (and their less than optimal repeatibility) of the various bacterial mutants.

3) Improve the stated justification of the work. The justification for the described work is framed around the enrichment of SNPs in MCPs of T1 strains and the potential that chemotaxis may have contributed to the expansion of the T1 lineage. One of the reviewers asked about evidence regarding selection. Moreover, counterintuitively, the manuscript focuses primarily on DC3000 (with T1 playing second fiddle in supplementary data), a lineage that T1 replaced. In addition to addressing the reviewer’s comment, I suggest the justification be reworded to better reflect the focus of the work. It may also behoove the authors to consider how similarities/differences in phenotypes of DC3000 and T1 mutants address the hypothesis that polymorphisms in chemotaxis-associated genes are associated with the success of T1 in tomato.

I also noticed that a data point for the wild type strain is missing in agar concentration 0.9% in figure 4.

Reviewer 1 ·

Basic reporting

I am surprised there are significant differences for Fig S3B bottom since the curves are so similar. There also do not appear to be error bars on Fig S3B. It would be easier to see any differences if you made the vertical scale cover a smaller range for Fig S3B bottom.

It is not clear if the plate images in Fig 4 and S4A were taken at the same time or from different experiments. I would recommend putting a white line to separate pictures of plates taken from different experiments to make this more clear. These images don’t appear to be evenly cropped. It would also look more consistent if the label was written consistently from left to right, or top to bottom for both panels and if the labels were aligned.

I think the mobile diameter would be more clear in Fig S4A if it was shown as in Fig 4 or S4B. The joined line implies a relationship among the agar concentrations.

The data presentation in Fig 5A and B is not as easy to interpret as in Fig S4. I would recommend that the authors switch to bar graphs.

There are some problems with reference formatting (line 77, 81, 129, 291, 555).

Line 126: This should be stratification if the cold period is used to break seed dormancy. Vernalization is cold treatment to promote flowering.
Line 131-133: There are several problems with this sentence.
Line 151-152: The end of this sentence doesn’t make sense to me.
Line 169: It would be clearer if you mention that cheA2- is the disruption strain in this section.
Line 172, 173: If this is one mutant, it should be “has a…”
Line 185, 198 (and possibly elsewhere as well): should be PtoDC3000
Line 204: neo should be italicized
Line 210: npt2 should be italicized
Line 239: should be adaptation
Line 251, 382: Gram-negative
Line 278-282: I think it would be simpler if you included both strains in your description of the phenotypes as they are the same. i.e. Pto1108cheA2- and PtoDC3000cheA2- exhibited a smooth-swimming… Pto1108cheA1- and PtoDC3000cheA1- swam and tumbled like the wild type Pto1108 or PtoDC3000 strains.
Line 321-323: This sentence doesn’t make much sense. Do you mean that the expression of these genes is thermo-regulated? If you are saying that these products are required for motility, then you don’t need to say “therefore motility” later in the sentence.
Line 390: identically
Line 430: the severity… was
Line 433: period missing
Table 1: strains are not lined up properly with the descriptions

Experimental design

It is puzzling that the disruption and deletion mutants don’t show the same phenotype, and that the deletion mutants cannot be complemented. The authors should confirm that the complemented strain produces cheA1 or cheA2.

The rationale for the hypothesis (line 364) that che2 is needed for effector delivery is not entirely clear. Do you mean that chemical sensing by cheA might direct the bacteria to the host? If so, I think it would be helpful to explain this a little more. Right now, it sounds like cheA is analogous to the T3SS in effector injection.

Validity of the findings

Line 236: The data shows there are SNPs. I did not see any tests for positive selection with these sequences. You cannot really conclude that there is positive selection if you haven’t tested for this.

It is concerning that the knock-out strains do not recapitulate the phenotype of the disruption mutants, and that the putatively complemented lines do not look wild type. I would suggest that the authors confirm that the disruption and knockout strain really do not express the RNA or protein (if tagged) of the interrupted gene. Is the expression of other genes in each cluster affected in the knock-out strains? It might help to include a schematic of the gene cluster organization.

Additional comments

The authors investigated the role of chemotaxis genes in pathogenicity and motility. They created disruption and knockout lines of two chemotaxis genes. cheA2 was necessary for motility at low temperature while cheA1 did not play a role in motility. cheA2 also contributed to bacterial growth in planta.

·

Basic reporting

Overall the reporting met the required standards, with a couple small exceptions:

A description of the minimal medium used should be included in the methods section.
A better description of plant sampling should be included in the methods, in particular whether endophytic or epiphytic populations (or both) were sampled following spray inoculation.

Experimental design

In general the experimental design is sound. However, using a diameter measurement for swarming is questionable. The swarming areas shown in the pictures are far from being being uniformly circular. It would be more appropriate to use an image analysis software to measure the swarming area.

Validity of the findings

The validity of plant inoculations are the biggest concern. See the 'general comments' for my specific concerns.

Additional comments

In this manuscript, Clarke and colleagues investigate the contribution of chemotaxis to various culture-based motility phenotypes as well as fitness or pathogenicity within host and non-host plant environments. They demonstrate the function of the che2 cluster in mediating flagellar-based chemotaxis (through a combination of phylogenetics, swimming motility assays, and a modified version of Adler’s split capillary tube assay), whereas the function of the che1 cluster remains unknown. In addition to describing the role of the che2 cluster in chemotaxis, the authors demonstrate an interesting temperature-dependent trade-off between flagellar-mediated surface motility (expressed under cool incubation temperatures) and pili-mediated surface motility (expressed under warm incubation temperatures). Finally, the authors present data indicating that potentially one or both of the che clusters are important for interactions between Pto strains and their host plants by comparing the populations the mutants with their wild type strains when spray inoculated onto either tomato or A. thaliana. Although this research is largely sound in execution and interpretation, there are a couple significant issues that should be addressed, as indicated below.

Major Comments:
1. The authors used two different approaches to disrupt the two che clusters: a disruption approach whereby either cheA1 or cheA2 are segmented into incomplete gene fragments (for both PtoDC3000 and Pto1108). Additionally, the authors use a deletion approach to remove the coding sequence for cheA1 and cheA2 from PtoDC3000 only. While the results across the cheA2 disruption strains (both PtoDC3K and Pto1108) and the cheA2 deletion strain in the PtoDC3K background are consistent in their phenotype (reduced apparent motility in swimming media and reduced chemotaxis in the capillary assay, figs 2 and S2), the authors are only able to convincingly complement the PtoDC3K deletion mutant (Fig. S2). The authors state that multiple independent cheA2 disruption mutants produce the same phenotype. All of the data is reasonably consistent to believe the in culture phenotype. It is less convincing, however, when looking at the plant inoculations. Did the authors use the both of the independently derived disruption mutants in these assays? If so, were the results consistent across the independent lines? Given the variability in the in planta phenotypes (see next comment), it is difficult to know whether the phenotypes are real or not.
2. The way the data is presented in fig 6A-D makes it appear highly ambiguous as to whether there is or is not an effect of the cheA1 or cheA2 mutations. In particular, no statistical analysis is presented. The authors present the number of times the experiment was performed that gave roughly similar results out of the total number of times the experiment was performed; this is an odd way to present the data. It would be better to perform an ANOVA across all experimental repeats (with each repeat treated as a blocking factor) to asses whether, given all of the data, there is reasonable support for a significant difference between the mutants and the wild type strains.
3. One additional factor complicating interpretation of in planta data is that it is unclear what fraction of the population is endophytic or epiphytic from the sampling scheme (which needs to be described better within the methods--hunting through a previous publication to determine for this ms whether the authors were sampling epiphytic, endophytic, or both populations is annoying). The authors state that “neither cheA1 nor cheA2 was consistently essential for pathogenicity when inoculated via infiltration…” but do not show the data. Given that the spray inoculations themselves were inconsistent to a degree, does this mean that infiltration was even less consistent that the spray inoculation? Or that the mutants were more consistently similar to the wild type strain in growth following infiltration compared to the spray inoculation? The infiltration data should be included. Distinguishing between the endophytic and epiphytic populations may help resolve some of the inconsistency of these results (either through surface sterilization or bath sonication), as well as sampling at earlier time points, or both. It is particularly difficult to interpret these results given the lack of effect for a fliC mutant (results from a previous publication that is cited, lines 415-416).

Minor Comments:
4. (Line 43) Remove extra parenthesis following Almeida citation.
5. (Line 46) ‘that’ should be ‘than’.
6. (Line 81) Scharf citation should not be italicized.
7. (Line 133) The MM should be defined in the methods section. It isn’t clear which MM is being used until the results section (line 291).
8. (Line 251) Gram should be capitalized in Gram-negative (also line 381).
9. (Line 707) The agar concentration range stated in the legend of fig. 4 differs from the concentration range stated in the figure iteslf.
10. (Line 726) This sentence needs to be edited “…leaves that should had a strong…”.

---

## Round 0.2 · Minor Revisions

Dear Chris,

Thank you for the rebuttal and new version of the manuscript.

I have attached a version, in which I have made a few changes to errors that I spotted (tracked) and added comments. These are meant to help improve the readability of the manuscript and had little effect on my decision regarding the manuscript. Please note that my comments are found in all sections of the manuscript.

The “minor revision” is with regards to my request for you to please work with your co-authors to try another attempt in addressing reviewer #2’s comment on the in planta assay. Because of the text starting on line 503, it is clear that the authors are aware of the limitations of the work and have thought of alternatives. But, I am not comfortable with how the experiment presented as figure 6E is framed; it comes across as too strong of a direct test of a specific mechanism. My reasons are as follows.

There is perceived disconnect between drawing the conclusion that chemotaxis primarily functions during the epiphytic phase (line 390) and suggesting that chemotaxis affects delivers of effectors (line 406), which is thought to occur primarily during the endophytic phase.

The implications of the proposed model that is being tested in figure 6E are not parsimonious:

1) cheA2 wild type secretes effectors (AvrRpt2 and HopAS1) that has cognate R genes in A. thaliana; mutation in cheA2 specifically dampens secretion of these effectors, while not negatively affecting the levels and functions of other effectors necessary for virulence.
2) cheA2 wild type secretes effectors that have cognate R genes in A. thaliana; mutation in cheA2 generally dampens secretion of all effectors to levels such that AvrRpt2 and HopAS1 are at levels below a threshold for triggering ETI. Yet despite the depression in secretion, virulence of the pathogen is not compromised.

Regardless of parsimony, the use of avrRpt2/HR does not test alternative models 1 or 2; this experiment only tests functionality of the T3SS.

There are no experiments that attempt to quantify changes in host resistance responses.

My suggestions:

1) I suggest you cite Vinatzer et al., (2006) and Lee et al., (2012), reports that suggested type III effectors influence epiphytic growth of bacteria.
2) Move 6E to supplemental and “casually” state in the text, that the T3SS system of cheA2 (and cheA1) mutants are still functional and can deliver type III effectors into plant cells.
3) Delete figure 6E and associated text, as an alternative to suggestion #2.

Best,

Jeff

---

## Round 0.3 · accepted · Accept

Thank you for your efforts in addressing the previous recommendations. I think the manuscript reads very well. As you can see, I have accepted this most recent version. However, as you work with the production staff, please fix the last errors. The most significant is that the HR assay is still described in the M&M section. I also spotted a double period on line 473. Please see the attached PDF; I simply indicated these areas with an empty comment box.

All the best,

Jeff